# Gait Recognition and Assistance Parameter Prediction Determination Based on Kinematic Information Measured by Inertial Measurement Units

**DOI:** 10.3390/bioengineering11030275

**Published:** 2024-03-13

**Authors:** Qian Xiang, Jiaxin Wang, Yong Liu, Shijie Guo, Lei Liu

**Affiliations:** 1Engineering Research Center of the Ministry of Education for Intelligent Rehabilitation Equipment and Detection Technologies, Hebei University of Technology, Tianjin 300401, China; 201811201005@stu.hebut.edu.cn (Q.X.); wangjx@hebut.edu.cn (J.W.); 202131205130@stu.hebut.edu.cn (Y.L.); 202131205073@stu.hebut.edu.cn (L.L.); 2The Hebei Key Laboratory of Robot Sensing and Human-Robot Interaction, Hebei University of Technology, Tianjin 300401, China; 3School of Mechanical Engineering, Hebei University of Technology, Tianjin 300401, China

**Keywords:** soft lower-limb exoskeleton, gait recognition, long short-term memory (LSTM), support vector machine (SVM), assistance parameter planning

## Abstract

The gait recognition of exoskeletons includes motion recognition and gait phase recognition under various road conditions. The recognition of gait phase is a prerequisite for predicting exoskeleton assistance time. The estimation of real-time assistance time is crucial for the safety and accurate control of lower-limb exoskeletons. To solve the problem of predicting exoskeleton assistance time, this paper proposes a gait recognition model based on inertial measurement units that combines the real-time motion state recognition of support vector machines and phase recognition of long short-term memory networks. A recognition validation experiment was conducted on 30 subjects to determine the reliability of the gait recognition model. The results showed that the accuracy of motion state and gait phase were 99.98% and 98.26%, respectively. Based on the proposed SVM-LSTM gait model, exoskeleton assistance time was predicted. A test was conducted on 10 subjects, and the results showed that using assistive therapy based on exercise status and gait stage can significantly improve gait movement and reduce metabolic costs by an average of more than 10%.

## 1. Introduction

Soft exoskeletons have been widely studied for their advantages of being lightweight, comfortable, and easy to wear and take off [1,2]. During walking, they obtain motion information through wearable sensors and apply assistance based on gait styles and phases estimated from the motion information. If the exoskeleton assistance does not match the intention of human movement, the exoskeleton will hinder human movement and even cause instability, leading to falls. Therefore, gait recognition is crucial for the control of a soft exoskeleton.

In terms of sensor system design, commonly used sensors include inertial sensors (IMUs) [3,4,5], plantar pressure sensors [6], surface electromyography electrodes [7,8], and electroencephalogram electrodes [9]. IMUs are widely used in gait motion recognition for their high portability, low cost, and environmental friendliness [10,11,12,13,14]. At present, many studies on gait recognition using different sensor configurations have been proposed. These studies are based on threshold methods for identifying gait events, which are predetermined based on experience. Because gait features are unique, each person’s threshold may vary. The threshold adjustment process is manual and time-consuming. With the development of statistical methods, various neural networks have been applied to gait recognition in an attempt to solve this problem.

There are two main types of exoskeleton recognition for human motion intention: one is the recognition of motion status under different road conditions, and the other is the recognition of gait phase. The motion state and gait phase are important bases for the time and magnitude of the auxiliary force applied by the exoskeleton [15]. By adjusting the timing and magnitude of the assistance force, the exoskeleton can match the gait characteristics of the wearer well, thereby improving adaptability and efficiency. However, the adjustment is usually based on experience, resulting in poor adaptability of the exoskeleton and poor auxiliary effects [16,17,18]. To solve this problem, human-in-the-loop (HIL) optimization [19,20], which is the method of optimizing control parameters for individualized assistance within soft exosuits, has attracted increasing attention. HIL optimization is an effective method for identifying optimal control parameters based on feedback from human gait information, and is particularly suitable for the personalized customization of auxiliary parameters for users [21,22,23]. The effectiveness of the HIL optimization strategy depends on the accuracy of human gait information recognition. At present, research mainly focuses on the accuracy of gait recognition and assistance prediction itself, with relatively little application in the control of lower-limb exoskeleton assistance force to analyze the application effect of recognition.

In order to predict the auxiliary time of an exoskeleton, we build a gait recognition model that can be applied to exoskeletons. The contributions of this paper are as follows: Firstly, a hip-assisted exoskeleton gait recognition method based on five inertial sensors is proposed, including SVM motion state recognition and LSTM phase recognition. Secondly, considering the coupling relationship between the two legs, the model proposes a rule-based gait event labeling method that can recognize both the left and right leg assist times. Finally, the gait recognition model is applied for the prediction of exoskeleton assisting time, and the effectiveness and reliability of the gait recognition model are proved by analyzing the assisting effect.

## 2. Related Works

In terms of motion state recognition, different recognition models were studied according to different applications, and certain recognition effects were achieved [24,25]. For example, Semwal et al. used a CNN to identify six different motion states [26], Weiland et al. used a classification artificial neural network (ANN) to recognize horizontal walking, uphill climbing, and uphill movement states [27], and Liu et al. placed four IMUs on the front of the thighs and the front of the calves of both legs and used an SVM to identify the current motion state [28]. The identifications in these studies are mostly typical of movement states, such as standing, horizontal walking, going up and down slopes, and going up and down stairs, while basic human movements include turning left and right, and so on. In addition, different motion states require different assistance planning techniques, and most of these motion recognition studies focus on recognition, and have not been well applied to the control of exoskeletons.

On the other hand, the phase recognition of the human gait is also a hot topic [29]. Zhang et al. proposed a multi-degree-of-freedom gait-detection method based on the kinematic information collected by IMUs on the dorsum of the foot, ankle, and thigh, and pursued the real-time recognition of the five phase features in a gait cycle [30]. Luo et al. identified the swinging and standing phases using a quadratic discriminant analysis classifier [31] based on the information of IMUs located on the thigh and shank. Pazar et al. proposed an LSTM model based on the information of IMUs and knee strains to recognize gait phases [32]. The above studies are focused on gait recognition itself; less attention has been paid to its application in the assistance force control of lower-limb exoskeletons.

There is an increasing amount of research on HIL. Koller et al. used a one-dimensional gradient descent method to determine the optimal assist parameters for ankle-assisted exoskeletons. At a walking speed of 1.2 m per second, the algorithm process for each subject took 50 min [33]. Kim et al. used the Covariance Matrix Adaptation Evolution Strategy (CMA-ES) to determine three optimal parameters: peak time, offset time, and force magnitude. Eight participants walked on the treadmill at a speed of 1.25 m/s, and the iterative optimization process took an average of two hours per participant. The research results indicate that metabolism significantly decreases under moderate intensity and optimal time [34]. Ye et al. proposed a Bayesian optimization algorithm that uses minimum metabolic cost as an indicator to conduct parameter optimization, minimizing the metabolic cost for humans walking on a 1.25 m/s treadmill [23]. The disadvantage of this method is that the devices used for optimizing parameters are not convenient to carry and travel with, so it is important to propose an assistance time recognition method based on a wearable sensor. In the above studies, the required time is relatively long and cannot meet the requirements of real-time online adjustment during walking.

## 3. Materials and Methods

This section will provide a detailed introduction to the proposed approach, including platform design, data acquisition and preprocessing, feature extraction, the gait recognition model, the experiment, and other modules.

### 3.1. The Soft Lower-Limb Exoskeleton Platform

We proposed a belt-type soft hip-assist exoskeleton, as shown in Figure 1. The exoskeleton consists of two actuation units, a control unit, a power supply unit, a waist brace, a knee brace, a pair of shoulder straps, and two belts for transmitting the assistant forces to the legs of the wearer. A load cell is fixed to each of the belts for measuring the assistant forces. The actuation units are tied to the front of the waist brace (each unit has a motor). The assist forces are transmitted by the belts from the actuation units to the knee joints, generating driving torques to the hip joint. Five IMUs are used to measure the motions of the lower limbs and the trunk of the wearer. The IMUs are 9-axis ones consisting of accelerometers, magnetometers, and gyroscopes. Only gyroscopes and accelerometers are used in this work, as magnetic sensors are susceptible to environmental influences. The placement diagram for the IMUs is shown in Figure 1. The IMUs are connected to the main control board inside the back shell of the exoskeleton by a long wire (transmission rate: 500 KHZ, sampling frequency: 200 Hz). All electronic devices of the exoskeleton are powered by a 0.55 kg power supply unit equipped with lithium-ion batteries (36 V, 3 Ah). The weight of the exoskeleton is approximately 2.7 kg.

The soft hip-assist exoskeleton we proposed provides hip flexion assistance force with two belts by imitating the function of the rectus femoris muscle, which is the main hip flexor in human walking. The force of the rectus femoris muscle has the form of Equation (1) [34]. Thus, it is reasonable for us to suppose that the profile of the assistive force takes a similar form. So, the assistance force is expressed as
(1)$$F=A\sin\left(\frac{\pi }{T_{a}}t+\alpha \sin\left(\frac{\pi }{T_{a}}t\right)\right)+f,$$
where F and A represent the assistance force and its magnitude; t represents time; α, the phase shift factor; $T_{a}$, the swing period; and f, the force offset to prevent the belts getting loose along the thigh of the swing leg without influencing hip extension.

Imitating the muscle force of the rectus femoris, we set the start time of the assistance force as the gait event of toe off, and the end time as the gait event of heel strike in a gait cycle. The time corresponding to the peak of the assistance is set as the gait event of hip max.

### 3.2. Data Collection and Preprocessing

We collected gait data from 25 healthy experimenters during standing (SD), level walking (LW), going up stairs (US), going down stairs (DS), going up slopes (USL), going down slopes (DSL), turning left (TL), turning right (TR), left steering (LS), and right steering (RS). Walking characteristics such as pace, stride length, and height were carried out in full accordance with the wishes of the experimenter for the whole walking process. After the exoskeleton powered on, the experimenter stood up first, and after one minute, began to complete 9 motion states: level walking, going up stairs, going down stairs, going up slopes, going down slopes, turning left, turning right, left steering, and right steering. Level walking is walking for approximately 100 m under the VICON motion capture system (Oxford Metrology Limited, Oxford, UK). Going up and down stairs involves walking the height of two floors in each direction. Walking up and down slopes occurs for about 50 m. The left and right turns were tested for 2 min, respectively, and 50 left and right turns were completed, respectively. Each of the 25 participants performed a complete experiment, and the entire experiment was performed 25 times. The details of the dataset are shown in Table 1. A total of 4,414,627 sample points were collected, including 299,922 SD samples, 497,373 LW samples, 595,492 US samples, 594,342 DS samples, 287,584 USL samples, 288,389 DSL samples, 584,770 TL samples, 583,571 TR samples, 343,127 LS samples, and 340,057 RS samples.

The human walk exhibits a periodic pattern known as a gait cycle [35]. Each gait cycle has a series of gait events that occur chronologically at specific time points, which are important for determining the starting point of applying assistance force. According to the exoskeleton assistance strategy, the gait times that need to be recognized during a gait cycle are left heel strike (L-HS), right toe off (R-TO), right hip max (R-HMax), right heel strike (R-HS), left toe off (L-TO), and left hip max (L-HMax) (Figure 2).

We selected a dataset of horizontal walking in various road conditions as the dataset for gait phase recognition. The verification of motion status is based on the division of each gait cycle by the VICON system. We used IMU information to identify gait events, but also used the VICON optical motion capture system to measure the motion of the human body during walking for calibration and verification. The IMU inertial system and the VICON system have their own hardware and software. They are synchronized based on velocity information. In the sagittal plane of the VICON system, the lowest vertical position of the heel is the HS event, the highest elevation of the toe from the lowest is the TO event, and the highest hip joint angle is the HMax event. The details of the dataset are shown in Table 2. A total of 497,373 sample points were collected, including 57,214 L-HS samples, 117,840 R-TO samples, 71,250 R-HMax samples, 56,790 R-HS samples, 119,590 L-TO samples, and 74,689 L-HMax samples.

### 3.3. Feature Extraction

In terms of input feature selection, features with different gait types are selected because, for example, the hip angle curve when descending stairs is different from other gaits, and the angle curve of the ankle joint differs significantly from other gaits when going uphill and downhill. The standard deviation of the combined acceleration of the center of mass is almost zero when people are at rest, but it changes greatly when they are moving. The center of mass acceleration standard deviation reflects the discrete path of the acceleration data.

The information from IMUs in different positions was used for the identification and comparison of the input feature quantities: 3 IMUs were based on the waist and foot, 3 IMUs were based on the waist and thigh, and 5 IMUs were based on the waist, foot, and thigh. After the experiment, it was found that the three IMUs of the waist and foot could not classify going up and going down. Based on the 3 IMUs on the waist and thigh, the recognition rate of level walking and turning was low (about 45%). Therefore, a total of 30 feature vectors, including the angular velocity and acceleration information of 5 IMUs of the waist, thigh, and foot, were used as input feature vectors for the gait recognition model.

For the feature extraction method of gait phase recognition, we consider the coupled relationship between legs, and propose a rule-based feature labeling method. HS: As shown in Figure 3A,D, when event HS occurs, the ankle reaches the maximum dorsiflexion angle. It can be detected by the zero-crossing detection rule. TO: As shown in Figure 3A,C,E, when event TO occurs, the ankle reaches the maximum rate of plantar flexion. It can be detected by a peak detection rule. HMax: As shown in Figure 3B,C,F, when event HMax occurs, the hip reaches the maximum flexion angle. It can be detected by the zero-crossing detection rule.

### 3.4. Gait Recognition Model

We use an SVM to recognize motion states from the motion information measured by the IMUs, as it requires fewer samples and exhibits strong generalization ability. A radial basis function (RBF) kernel is used to transform the low dimensional data into a high-dimensional space [36], and a gait classification hyperplane is constructed in the high-dimensional space. This hyperplane can separate non-linear and non-separable data into linearly separable data. The classification hyperplane is defined as
(2)$$f(x)=\omega^{T}\varphi (x)+b,$$
in which ω is the hyperplane normal vector, φ(x) is the input feature vector, and b is the intercept.

Then, gait classification can be transformed into a problem of finding the maximum distance between walking styles, and this distance is expressed as
(3)$$\max\mathrm{dis}=\underset{w}{\max}\;2\times \frac{y(\omega^{T}\varphi (x)+b)}{{\left\|w\right\|}^{2}},$$
where $y_{i}(\omega^{T}\varphi (x)+b)=1,\forall y_{i}\in \left\{-1,1\right\}$. Thus, the problem of solving the optimal gait classification surface via SVM can be reduced to a quadratic programming problem, as shown by Equation (4). Introduce slack variables ξ to transform the inequality constraints in Equation (4) to an equality constraint, as shown in Equation (5). We can build a Lagrange function, as shown in Equation (6).
(4)$$\left\{\begin{array}{c} \min dis=\underset{w}{min}\frac{{\left\|w\right\|}^{2}}{2}\\ \left\|w\right\|=\sqrt{{w_{1}}^{2}+{w_{2}}^{2}+\cdots +{w_{n}}^{2}}\\ y(\omega^{T}\varphi (x)+b)\geq 1,\forall i=1,2,\cdots ,n \end{array}\right.,$$
(5)$$h_{i}(w,\xi_{i})=1-y_{i}(w^{T}\varphi (x)+b)+\xi_{i}$$
(6)$$L(w,b,\xi ,\lambda ,\mu )=\frac{1}{2}{\left\|w\right\|}^{2}+C\sum_{i=1}^{m}\xi_{i}+\sum_{i=1}^{n}\left(\lambda_{i}h_{i}-\mu_{i}\xi_{i}\right)$$
where λ and μ are Lagrange multipliers, w and b are optimization parameters, and C represents the penalty factor. Taking the derivatives of the optimization parameters in Equation (6), we obtain the gait classification hyperplane as
(7)$$f(x)=sign(\sum_{i=1}^{N}w_{i}\xi_{i}K(x_{i},x)+b_{i})$$

By training the model using a multi-classification method with binary tree structures, we can recognize the walking styles. Figure 4 gives the framework of the recognition. The data collected by the IMUs are the accelerations and angular velocities of the waist and right and left legs, represented by $\left[a_{fr},v_{fr},a_{fl},v_{fl},a_{tr},v_{tr},a_{tl},v_{tl},a_{w},v_{w}\right]$. The penalty factor C is 2. Choose Gaussian (RBF) as the kernel function based on the characteristics of the data, and gamma is set to scale. It automatically calculates the value of gamma based on the scale of the input data.

We take the kinematic data of walking motion as a periodic time series and employ long short-term memory (LSTM) to identify gait phase. The cell state of the LSTM allows for the retention or addition of input information to the cell state, enabling a better integration of historical information for the recognition of gait phase at the current time point [9,37,38]. The LSTM cell consists of oblivion gate $f_{t}$, input gate $i_{t}$, output gate $o_{t}$, and cell state $C_{t}$.
(8)$$f_{t}=\sigma (W_{f}\cdot \left[h_{t-1},x_{t}\right]+b_{f})$$
(9)$$i_{t}=\sigma (W_{i}\cdot \left[h_{t-1},x_{t}\right]+b_{i})$$
(10)$$o_{t}=\sigma (W_{o}\cdot \left[h_{t-1},x_{t}\right]+b_{o})$$
(11)$$C_{t}=f_{t}\cdot C_{t-1}+i_{t}\cdot \tanh(W_{c}\cdot \left[h_{t-1},x_{t}\right]+b_{c})$$
(12)$$h_{t}=o_{t}\cdot \tanh(C_{t})$$
where σ is the sigmoid function, W represents the weight matrices of each gate, and b, the bias vectors of each gate.

The network model used for gait phase recognition is shown in Figure 2. The input to the network is the time series $X=\left[x_{1},x_{2},\cdots ,x_{n-1},x_{n}\right]$, where $x_{n}$ is the vector consisting of the standardized IMU inertial raw data and n is the sequence length, which is 5. The epochs of the model are 200, and the number of samples in each training batch is 64. The gait phase recognition model consists of an LSTM layer and a fully connected (FC) layer followed by softmax. The input feature dimension is 30, the LSTM layer has 128 neurons, and the FC layer has 32 neurons. The output represents the probabilities of 6 different gait phases. This model runs on a notebook PC. The notebook PC receives the gait data sent by the robot, uses a trained LSTM model to obtain gait classification results, and finally sends the results to the exoskeleton through Wi-Fi. The artificial neural network training is implemented using Python 3.7.1 and TensorFlow 2.5.0.

## 4. Experiment

### 4.1. Subjects

The dataset for the gait recognition experiments was obtained from 25 healthy adults (age = 27.5 ± 2.4 years, height = 173.6 ± 5.9 cm, and weight = 69.5 ± 3.6 kg). The experiment was approved by the Ethics Committee of Hebei University of Technology, and each subject was asked to read and provide written informed consent before the test (Ethics Committee name: the Biomedical Ethics Committee of Hebei University of Technology; approval code: HEBUThMEC2023017).

### 4.2. Protocol

#### 4.2.1. Gait Classification and Gait Phase Recognition

For the recognition verification of multiple road conditions, the data of the first 20 people in the dataset in Table 1 were used to train the SVM model, and the data of the last 5 people were used to test the performance of the model. For gait phase recognition validation, the data of the first 22 individuals in the dataset in Table 2 were used to train the LSTM model, while the data of the last 3 individuals were used to test the performance of the model. In this paper, we use accuracy (ACC), $F_{1}$ scores, and the Matthews correlation coefficient (MCC) to evaluate the performance of the model. Different recognition models were trained offline using Python 3.7.1 and TensorFlow 2.5.0 software. The performance of the motion state and gait phase recognition algorithm was validated using a five-fold cross validation.

#### 4.2.2. Assistance Effect Experiment

Ten subjects were randomly selected from the twenty-five subjects to test the assistance effect of the assist policy based on the proposed gait classification and gait phase recognition. The exoskeleton’s assist time predicts the time difference in the gait event from the previous step of gait recognition. In order to verify the effectiveness of the gait recognition model, a metabolic power experiment for the exoskeleton-assisted effect experiment based on BP neural network gait phase recognition was added.

Each subject wore specially designed sportswear under the soft exoskeleton so that reflective markers could be pasted to them for the VICON system (Oxford Metrology Limited, Oxford, UK) to capture walking motions. Figure 5A presents the locations of the 16 markers in total that were pasted symmetrically to the left and right of the human body [39,40]. Each subject participated in two different speed walking experiments on a treadmill. Before the test, each subject was instructed to familiarize themselves with treadmill walking. To confirm the effect of the assist, power-off (P-OFF) tests were also carried out before the test with the assist (P-ON) for each walking speed. Figure 6 gives the test conditions.

Besides measuring the kinematic parameters in the assisted walking test, change in metabolic cost was also estimated using a portable respiratory measurement device (Cosmed, K4b2, Rome, Italy) worn by the subjects (Figure 5B). All the subjects were asked not to consume food or beverages 2 h before the test so that the metabolic cost could be estimated accurately.

### 4.3. Data Collection

Data were collected at 100 Hz using the VICON Motion system (Oxford Metrics, Oxford, UK) with 10 cameras (model: VANTAGE-V5-VS-5299). Real marker trajectory data were filtered with a quintic spline filter based on the code by Herman Woltring [41]. The position information of each subject’s joints was directly obtained. The angles of the hip, knee, and ankle joints were obtained using the plug-in gait model in Vicon Nexus software v1.8.5 [42,43,44]. All the joint angles were calculated from the YXZ Cardan angles, derived by comparing the relative orientations of the two segments [42]. Using Vicon Polygon (Oxford Metrics Group, Oxford, UK) [42] software, kinematic data were extracted to a C3D file or an ASCII file, which was then input into MATLAB software (MATLAB 23.2.0) for post-processing.

The respiratory data, oxygen consumption ($V_{O_{2}}$ [L/min]) data, and respiratory quotient (RER) were averaged across the latter half of the one-minute intervals for each sampling parameter condition. The net metabolic cost was calculated by subtracting the resting metabolic cost from the walking metabolic cost, and then normalized according to the weight of each subject.

### 4.4. Statistical Analysis

Means and standard deviations for each test condition were calculated. Statistical analysis was performed using the SPSS statistical software system (SPSS Inc., Chicago, IL, USA; version 22.0). One-way repeated-measures analyses of variance with three conditions (No-Exo, P-Off, P-On) were used to verify the effect of the hip assistance on metabolic reduction. Pairwise comparisons with Bonferroni post hoc tests were conducted to identify the differences between conditions when a statistically significant main effect was identified with the one-way repeated-measures analyses of variance. A paired *t*-test was performed to assess the difference between the different conditions for metabolic reduction. *p* < 0.05 represented a significant difference.

## 5. Results

### 5.1. Accuracy of Motion State Recognition

We compared the accuracy of four motion recognition models—KNN, Random Forest, Decision Tree, and SVM—as shown in Table 3. It can be seen that SVM has the highest recognition rate of 99.98%. Table 4 shows the confusion matrix of the motion state recognition, which shows the recognition results of the ten walking styles. It can be seen from Table 4 that a small portion of standing periods were identified as steering. The reason is that there were a few similar features between the two walking styles. The recognition rate of level walking was high. This is worth noting because we are more interested in assisting in level walking for frail, older individuals, and thus in removing assistance for other walking styles for safety.

### 5.2. Accuracy of Gait Phase Recognition

Table 5 compares the recognition rates of the different gait phases in a gait cycle. Table 3 shows the performance indicators of the different phase recognition models. It can be seen that, compared with NN, RBF and BP, LSTM has the highest recognition rate, which is 98.26%. The confusion matrix displays the recognition results of six gait phases, where the sample between two consecutive gait events is marked as the starting event. From Table 6 and Figure 7, it can be seen that the recognition rate of HS gait events (L-HS: 0.975, R-HS: 0.970) is lower than those of TO (L-TO: 0.991, R-TO: 0.995) and HMax (L-Hax: 0.983, R-Hax: 0.991) events. The reason is that the feature used to mark HS gait events was the zero-crossing detection rule of the Z-axis angular velocity of the foot IMU. Before the HS event, the Z-axis angular velocity of the foot IMU rapidly decreased in a short time. There was a certain error in finding the zero point during this rapid descent process. Although the recognition rate of HS was relatively low, the recognition accuracy was sufficient to apply to the assistance of the exoskeleton.

### 5.3. Kinematic Effect of the Assistance

The accuracy of gait recognition and the prediction of assistance parameters directly affects the effectiveness of wearing exoskeleton assistance, and the direct representation of this on the human body is the changes in the joints of the lower limbs during walking. The highest hip joint swing angle value during exoskeleton-assisted walking was significantly higher than that without exoskeleton walking. At speeds of 3.6 km/h and 5.5 km/h, the maximum swing angle of the hip joint increased by 5–6 degrees (Figure 8A,B). This indicates that setting the peak assist at the moment of maximum hip joint swing angle is effective and reasonable. Exoskeleton assistance generates torque on the hip joint. Although there is no torque generated on the knee joint, the force of the exoskeleton acts on the knee joint, and the angle of the knee joint also changes significantly. As shown in Figure 8E,F, at speeds of 3.6 km/h and 5.5 km/h, wearing an exoskeleton for assistance increases the maximum knee joint swing angle by 9 degrees and 7 degrees, respectively, compared to not wearing an exoskeleton.

As shown in Figure 8G–I, in toe off (point F, Figure 8I), there was a significant change in ankle plantar flexion, with an increase of 3–5 degrees. At the end of the assistance, there was a significant change in ankle dorsiflexion and an increase of 3 degrees in the angle (point E, Figure 8H). The changes in ankle joint plantar flexion and dorsiflexion corresponded to toe rocker and heel rocker, with an increased angle indicating better rolling effect and a better performance of human ankle rolling. This indicates that starting assistance at TO and ending assistance at HS is effective and reasonable. The changes in ankle joint plantar flexion and dorsiflexion can affect the ground clearance position of the heel and toe, and can be used to further analyze the changes in foot ground clearance distance. The position of the heel and toe (Figure 9A,C,D) and the position of the toe (Figure 9A,B) when the foot swings to the neutral position are both elevated under the assistance of exoskeletons. The elevation of the heel and toe positions reduces the risk of tripping over obstacles, which is one of the main purposes of hip joint assistance.

The combined result of changes in the swing angles of the human hip joint, knee, and lower limbs is an increase in step size, as shown in Figure 10. At two different walking speeds, there was an increase of 5–10 cm under assisted conditions. At the walking speed of 5.5 km/h, the step length increases even more, by about 10 cm. When walking speed is high, the more obvious the assist effect.

### 5.4. Metabolic Effect of Assistance

Figure 11 shows the average metabolic cost of ten subjects walking on a treadmill at a walking speed of 3.6 km/h and 5.5 km/h. Compared to not wearing an exoskeleton, wearing an exoskeleton at two different walking speeds reduced the metabolic power of the ten individuals by 10.98% (0.405 w/kg) and 14.28% (0.621 w/kg), respectively. The higher the speed, the more the metabolic power decreased, indicating that the assistance effect of exoskeletons is significant. As can be seen from Figure 11, the metabolic power of gait phase recognition based on a BP neural network for assist time prediction decreased by 6.3% (0.213 w/kg) and 8%,2% (0.357 w/kg), respectively. The metabolic power reduction in the SVM and LSTM recognition models proposed in this paper was significantly more than in the BP neural network, indicating that the proposed gait recognition model can be applied to exoskeleton-assisted control. This demonstrates the effect of the gait event based on assistance policies. When the respiratory exchange rate (RER) is less than 1, the subject is in an aerobic breathing state. When RER ≥ 1, the subject is in a respiratory state in which aerobic and anaerobic respiration work together. An increase in RER to some extent reflects an increase in exercise intensity and changes in the respiratory substrates of the body’s energy supply system. The RER of the ten experimenters was less than 1 under all three conditions, and the RER was the lowest under the assistance condition.

## 6. Discussion

With regard to kinematic analysis, there are still some issues worth discussing. We believe the slight improvement in point B (Figure 8A,C) is the result of alternating leg assistance. When the left leg is at the hip joint angle feature, point B and has not yet entered the assist phase, and the right leg has just completed the swing phase. Before the left leg enters the swing phase, potential energy is converted into kinetic energy to drive the movement of the left leg. Therefore, although assistance has not yet begun at point B, due to the conversion between kinetic energy and potential energy, the hip joint angle will also increase.

It should also be noted that, in the test, with the assistance of the exoskeleton, the step length significantly increased. Because the test was conducted at a constant speed on a treadmill, as the step length increased, the cadence inevitably decreased. Another possibility is that the human body autonomously changes its walking style, increasing step length and reducing step frequency.

When walking at asynchronous speeds, the balance system of the human body adjusts the relationship between step length and cadence to ensure an efficient walk, i.e., from the viewpoint of an efficient walk, an optimum relationship exists between step length and cadence. This is similar to the walking ratio (the ratio of step length to cadence) proposed by Murakami [45,46]. That is why the concept of walk ratio is usually used when walk efficiency is discussed. The metabolic consumption test demonstrated that the assistance did bring metabolic reduction. From the metabolic consumption test, it can be seen that the wearer’s metabolic cost did indeed decrease under assist conditions, ruling out the possibility that the increase in step length may not be the assisting effect.

At present, we have identified and perceived the gait phase of horizontal walking based on an assistance strategy for horizontal walking. However, there are significant differences in the walking characteristics of the human body under different road conditions. For example, in uphill and downhill movements, the main movement of the lower-limb hip joints is flexion, and the flexion angle is relatively small compared to horizontal walking as a whole. During the process of going up stairs, the flexion angle of the hip joint varies greatly. The assistance strategy of exoskeletons varies under different road conditions, and the effective implementation of assistance strategies requires an accurate perception of human motion intentions. Therefore, our plan for our next work is to study the perception and assistance strategies of gait phase under different road conditions.

## 7. Conclusions

Firstly, this paper proposed a gait recognition method using the information measured by five IMUs fixed on a soft exoskeleton. It distinguishes walk styles with the algorithm of a support vector machine (SVM) and gait phases with long and short-term memory (LSTM). Tests on 12 subjects demonstrated high accuracies of 99.98% and 98.06%, respectively. Secondly, a method for determining assistance parameters for the soft exoskeleton based on gait event and gait phase was proposed. Tests were conducted on three subjects to confirm its effect. The results showed that the assistance could significantly improve gait motion and reduce metabolic cost by an average of over 10%.

## Figures and Tables

**Figure 1 bioengineering-11-00275-f001:**
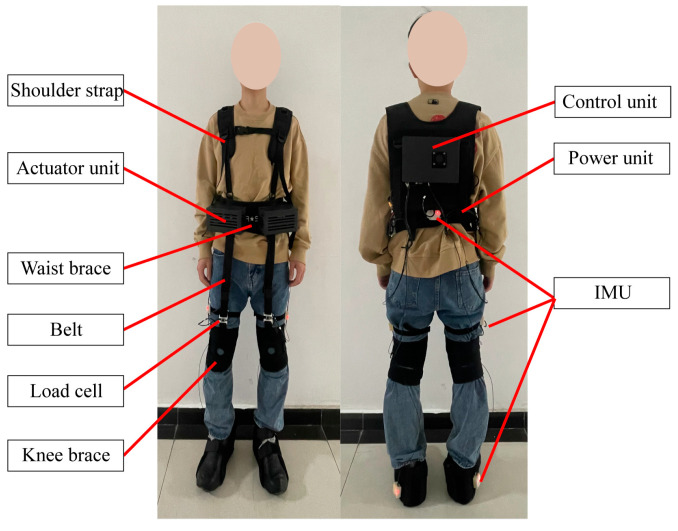
The soft lower-limb exoskeleton.

**Figure 2 bioengineering-11-00275-f002:**
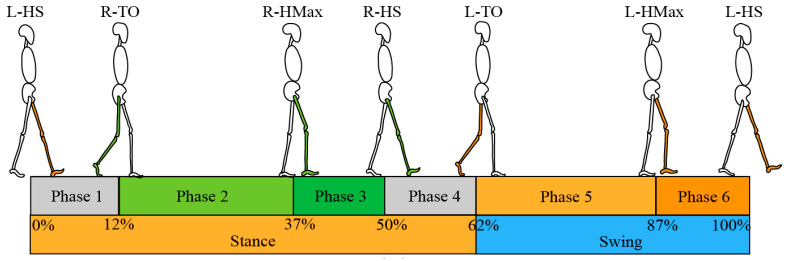
Gait event division within a gait cycle.

**Figure 3 bioengineering-11-00275-f003:**
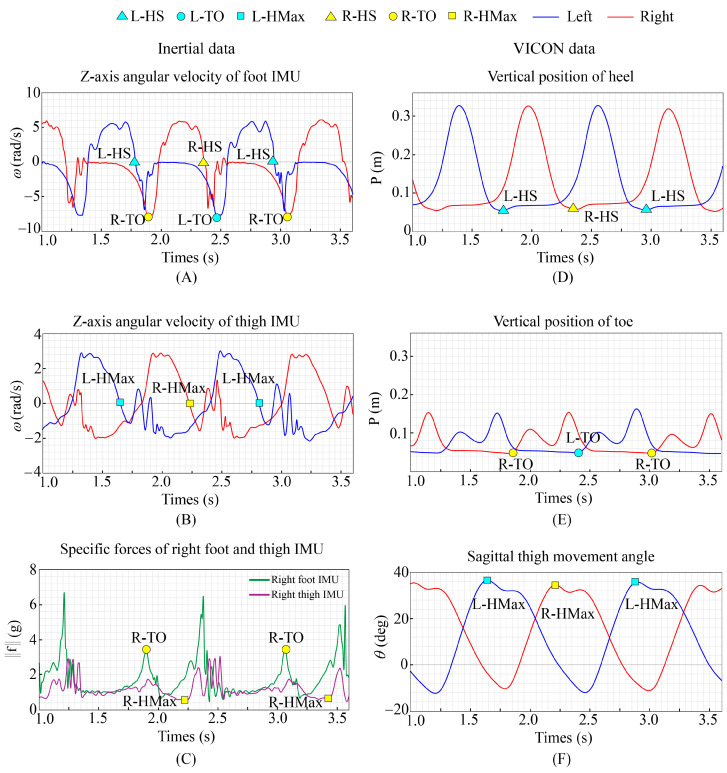
(**A**) The angular velocity of the Z-axis of the IMU located on the foot. (**B**) The angular velocity of the Z-axis of the IMU located on the thigh. (**C**) The specific force of the Z-axis of the IMU located on the right foot and thigh. (**D**) The vertical position of the heel obtained from VICON. (**E**) The vertical position of the toe obtained from VICON. (**F**) The hip angle obtained from VICON.

**Figure 4 bioengineering-11-00275-f004:**
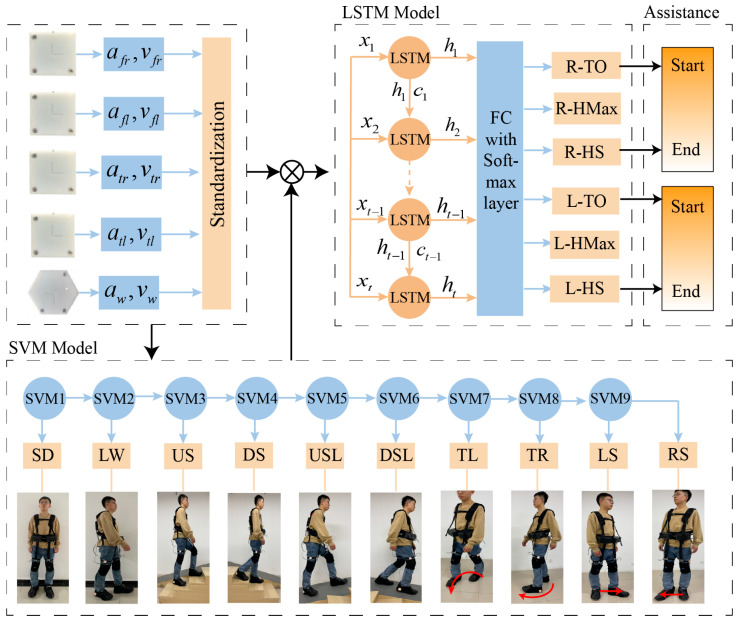
The gait recognition framework. The red curve represents the turning, with the arrow direction clockwise indicating the turning left (TL) and counter clockwise indicating the turning right (TR); The red straight line represents the steering, with the arrow direction to the right indicating the left steering (LS) and the direction to the left indicating the right steering (RS).

**Figure 5 bioengineering-11-00275-f005:**
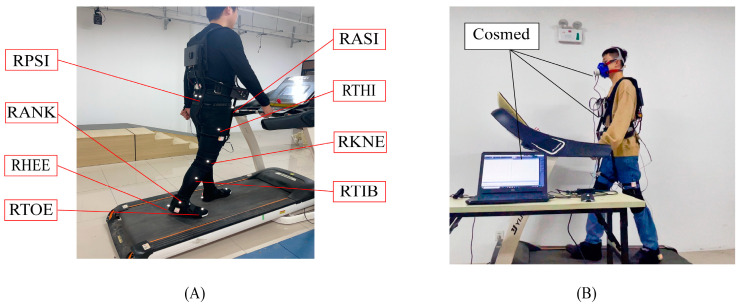
Testing assistance effect. (**A**) Kinematic test. (**B**) Metabolic test.

**Figure 6 bioengineering-11-00275-f006:**
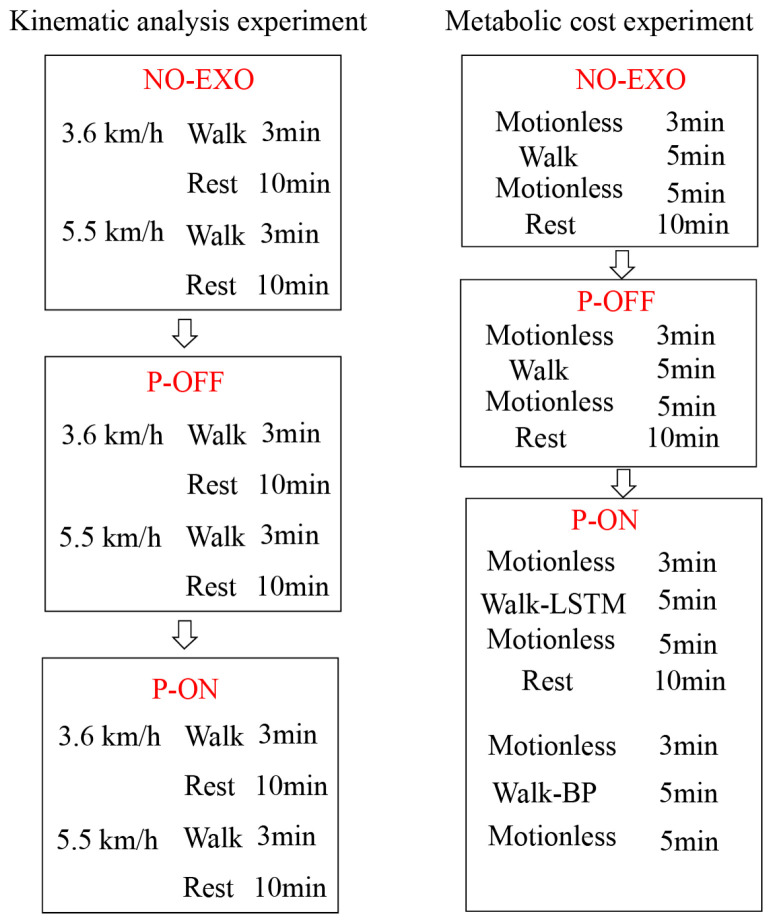
Test conditions.

**Figure 7 bioengineering-11-00275-f007:**
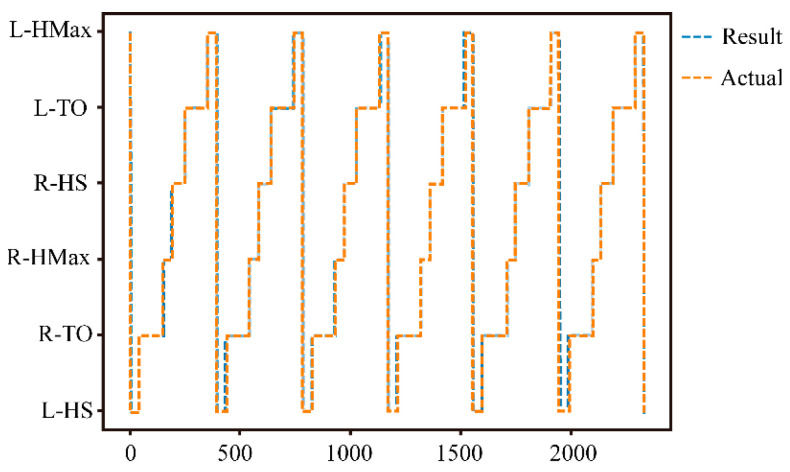
Actual data and recognition results for six gait cycles.

**Figure 8 bioengineering-11-00275-f008:**
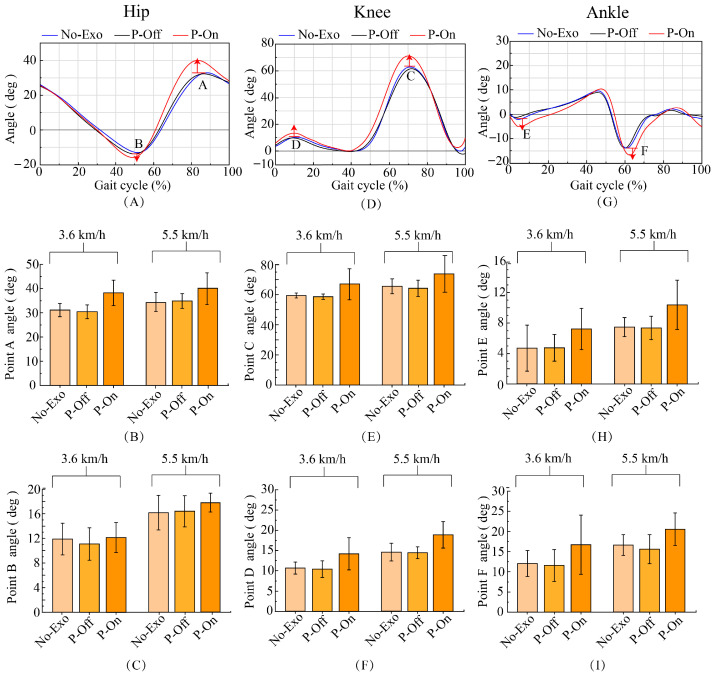
(**A**) The typical gait characteristics of the hip angle. Point A represents the typical point with the highest hip angle and point B represents the typical point with the lowest hip angle. (**B**) Changes in point A angles of the hip joint. (**C**) Changes in point B angles of the hip joint. (**D**) The typical gait characteristics of the knee angle. Point C represents the typical point with the highest knee angle during the swing period and point D represents the typical point with the highest knee angle during the support period. (**E**) Changes in point C angles of the hip joint. (**F**) Changes in point D angles of the hip joint. (**G**) The typical gait characteristics of the ankle angle. Point E represents the typical point with the lowest ankle angle and point F represents the typical point with the lowest ankle angle. (**H**) Changes in point E angles of the ankle joint. (**I**) Changes in point F angles of the ankle joint.

**Figure 9 bioengineering-11-00275-f009:**
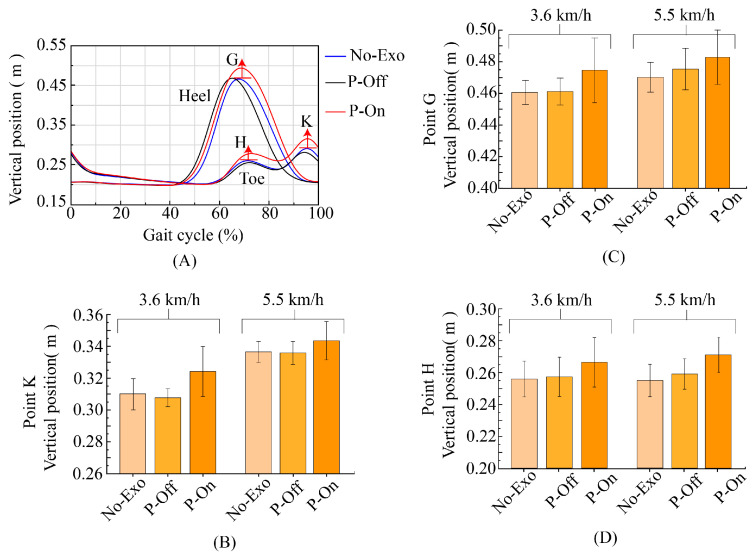
(**A**) The typical gait characteristics of the heel and toe during a gait cycle. Point G represents the highest point of the vertical position of the heel during the swing period. Point H and point K respectively represent the highest point of the vertical position of the toes during the swing period and before the foot lands on the ground. (**B**) Vertical position of point K at the toe under different conditions. (**C**) Vertical position of point G at the heel under different conditions. (**D**) Vertical position of point H at the toe under different conditions.

**Figure 10 bioengineering-11-00275-f010:**
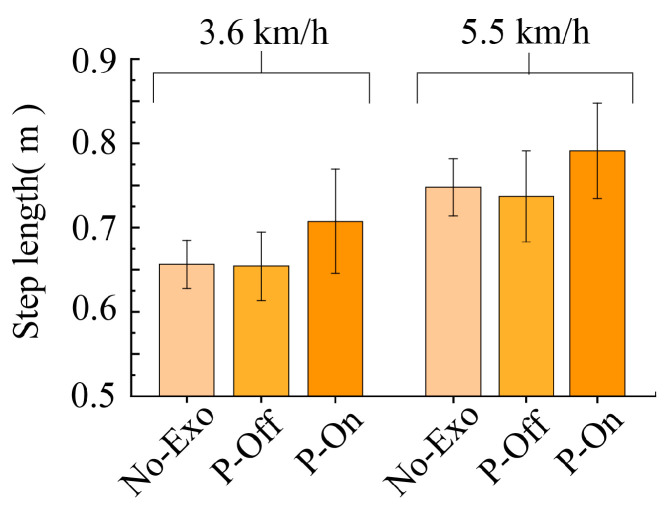
Changes in step length under different conditions.

**Figure 11 bioengineering-11-00275-f011:**
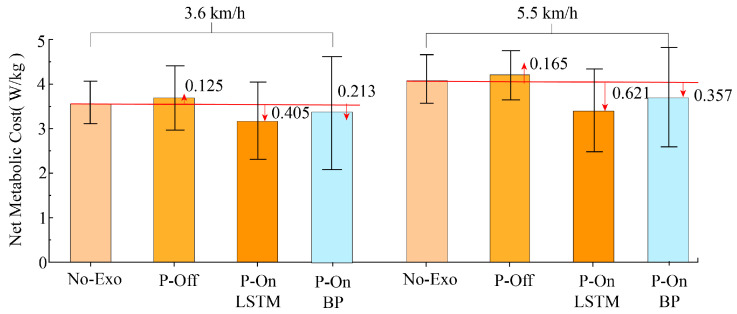
Changes in metabolic cost under different conditions.

**Table 1 bioengineering-11-00275-t001:** Dataset for different road conditions.

Trials	SD	LW	US	DS	USL	DSL	TL	TR	LS	RS
1	12,000	19,850	23,768	23,819	11,685	11,625	23,584	22,926	13,835	12,244
2	12,000	19,760	23,818	23,862	11,323	11,347	23,691	23,768	12,847	13,745
3	11,998	19,863	23,633	23,784	11,569	11,572	23,396	23,294	14,354	12,961
4	11,999	19,968	23,897	23,990	11,353	11,190	23,281	22,543	14,242	14,485
5	12,000	19,852	23,778	23,778	11,992	11,670	23,030	23,849	13,372	13,656
6	11,994	19,931	23,895	23,692	11,509	11,309	23,577	22,741	14,561	13,304
7	12,000	19,931	23,951	23,586	11,776	11,740	23,888	23,645	13,112	14,472
8	12,002	19,803	23,743	23,541	11,456	11,466	23,132	23,560	12,997	14,447
9	12,000	19,984	23,953	23,900	11,187	11,270	23,503	23,865	12,783	12,652
10	11,986	19,818	23,845	23,596	11,178	11,859	23,701	22,740	14,815	14,413
11	12,000	19,872	23,788	23,898	11,912	11,620	23,942	23,899	12,435	12,133
12	11,985	19,900	23,934	23,733	11,238	11,552	23,852	23,308	14,132	14,659
13	12,000	19,914	23,869	23,671	11,123	11,392	22,737	22,588	12,579	13,929
14	11,995	19,940	23,793	23,835	11,607	11,929	23,176	22,976	13,502	12,503
15	11,998	19,807	23,811	23,889	11,255	11,973	23,062	23,569	14,577	14,488
16	12,000	19,863	23,591	23,760	11,928	11,356	23,924	22,872	13,954	14,751
17	12,000	19,945	23,851	23,579	11,689	11,244	23,040	23,845	12,975	13,747
18	11,997	19,948	23,916	23,851	11,524	11,278	23,060	23,926	12,986	12,885
19	12,002	19,894	23,655	23,951	11,279	11,721	23,052	22,989	13,762	14,419
20	12,000	19,937	23,889	23,916	11,807	11,199	22,793	23,953	14,409	12,288
21	11,986	19,889	23,500	23,553	11,113	11,823	23,767	23,397	13,892	13,534
22	12,000	19,975	23,982	23,678	11,873	11,216	23,679	23,269	13,458	14,566
23	11,985	19,889	23,693	23,799	11,595	11,769	22,976	23,759	14,776	13,311
24	12,000	19,971	24,012	23,701	11,224	11,435	22,951	22,834	14,380	13,768
25	11,995	19,869	23,927	23,980	11,389	11,834	23,976	23,456	14,392	12,697
Total	299,922	497,373	595,492	594,342	287,584	288,389	584,770	583,571	343,127	340,057

**Table 2 bioengineering-11-00275-t002:** Dataset of gait phase recognition for horizontal walking.

Trials	LW	L-HS	R-TO	R-HMax	R-HS	L-TO	L-HMax
1	19,850	2504	4817	2981	2202	4615	2731
2	19,760	2129	4702	2769	2358	4876	2926
3	19,863	2391	4676	2783	2091	4895	3027
4	19,968	2213	4956	2796	2315	4983	2705
5	19,852	2318	4484	2980	2250	4640	3180
6	19,931	2242	4465	2791	2380	5054	2999
7	19,931	2457	5060	2791	2271	4952	2400
8	19,803	2220	4453	2874	2219	4951	3086
9	19,984	2188	4599	2997	2166	4953	3081
10	19,818	2367	4527	2776	2444	5059	2645
11	19,872	2466	5084	2783	2208	4578	2753
12	19,900	2044	4756	2787	2387	4898	3028
13	19,914	2117	4798	3088	2258	4634	3019
14	19,940	2440	4502	2892	2515	4657	2934
15	19,807	2355	4572	2874	2202	4976	2828
16	19,863	2048	4592	2782	2377	4873	3191
17	19,945	2311	4644	2792	2113	4854	3231
18	19,948	2199	5023	2793	2156	4603	3174
19	19,894	2597	4627	2786	2299	4863	2722
20	19,937	2290	4622	3091	2142	4664	3128
21	19,889	2082	4790	2785	2367	4592	3273
22	19,975	2459	5077	2696	2155	4618	2970
23	19,889	2043	4439	2785	2211	4637	3774
24	19,971	2280	5067	2896	2321	4586	2821
25	19,869	2454	4508	2882	2383	4579	3063
Total	497,373	57,214	117,840	71,250	56,790	119,590	74,689

**Table 3 bioengineering-11-00275-t003:** Evaluation of motion state recognition models.

Method Model	ACC ^1^ (%)	$F_{1}$ Score	MCC
KNN	98.72	0.9878	0.9880
Random Forest	99.31	0.9942	0.9935
Decision Tree	98.95	0.9894	0.9901
SVM	99.98	0.9985	0.9987

^1^ Accuracy refers to correctly determined motion state.

**Table 4 bioengineering-11-00275-t004:** Confusion matrix of the trained SVM.

	SD	LW	US	DS	USL	DSL	TL	TR	LS	RS
SD	0.99	0	0	0	0	0	0	0	0	0
LW	0	1.00	0	0	0	0	0	0	0	0
US	0	0	1.00	0	0	0	0	0	0	0
DS	0	0	0	1.00	0	0	0	0	0	0
USL	0	0	0	0	1.00	0	0	0	0	0
DSL	0	0	0	0	0	1.00	0	0	0	0
TL	0	0	0	0	0	0	1.00	0	0	0
TR	0	0	0	0	0	0	0	1.00	0	0
LS	0.01	0	0	0	0	0	0	0	0.99	0
RS	0	0	0	0	0	0	0	0	0.01	1.00

**Table 5 bioengineering-11-00275-t005:** Evaluation of gait phase recognition models.

Method Model	ACC ^1^ (%)	$F_{1}$ Score	MCC
NN	96.36	0.9635	0.9612
RBF	96.78	0.9675	0.9680
BP	97.55	0.9752	0.9748
LSTM	98.26	0.9821	0.9825

^1^ Accuracy refers to correctly determined gait phase.

**Table 6 bioengineering-11-00275-t006:** Confusion matrix of the trained LSTM.

	L-HS	R-TO	R-HMax	R-HS	L-TO	L-HMax
L-HS	0.975	0	0	0	0	0.017
R-TO	0.010	0.995	0	0	0	0
R-HMax	0	0.005	0.991	0.010	0	0
R-HS	0	0	0.009	0.970	0	0
L-TO	0	0	0	0.020	0.991	0
L-HMax	0.015	0	0	0	0.009	0.983

## Data Availability

The original data are available following reasonable request.

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
