# Peer review of "Gait Recognition and Assistance Parameter Prediction Determination Based on Kinematic Information Measured by Inertial Measurement Units"

_bioengineering, 2024, doi:10.3390/bioengineering11030275_

Round 1

Reviewer 1 Report

Comments and Suggestions for Authors

The manuscript under review presents a gait recognition method by using the information measured by five IMUs fixed on the soft exoskeleton and also proposes a dataset. The paper is easy to follow, however, the reviewer has the following recommendations.

- The technical novelty of the method is limited, it simply employs existing off-the-shelf components to recognize gait using IMU data. The contributions in terms of technical advancements should also be presented in the introduction section.

- The gait analysis using IMU data has recently received significant research, some traces can be seen in the related work too. However, the performance of the presented method is not compared with any of the existing methods. A comparison with the current state-of-the-art (SOTA) would further highlight the effectiveness of the presented method.

- The dataset proposed in this research appears to be not enough to claim the performance gains reported in the paper. More data should be collected, maybe in a different environment to test the robustness of the system.

- The effectiveness of the method is claimed on tests conducted on just three subjects. This is insufficient. The evaluation is not comprehensive. More experiments should be conducted.

- Lastly, wearing all sensors and carrying a system weighing around 3 kg might not be practical for elderly and sick people.

Reviewer 2 Report

Comments and Suggestions for Authors

The authors proposed an approach to assess gait recognition performance of an exoskeleton using machine learning approaches. Here are my comments:

-          Problem definition can be extended in abstract.

-          Problem definition and background needs more detail in intro section.

-          A separate related work section is needed.

-          Last paragraph seems to be incomplete. Please clearly indicate your approach.

-          Organizational paragraph is missing.

-          How did the authors use SVM as classifier this information is missing in the text.

-          How did the authors validate these motion states?

-          2.1 should be dataset and dataset descriptions should be given there.

-          How many channels did the authors use? They should also do a channel wise comparison of methods.

-          A separate feature extraction section is needed.

-          Class distributions should be given in a table.

-          A separate section that explains experimental setup is needed. This section should include evaluation criteria, train/test setup and model parameters.

-          What is the sample size for train and test?

-          How did the authors decide ML model parameters?

-          More evaluation metrics can be added, for example MCC value.

-          What are the parameters of LSTM model?

-          Also, brief explanations of the ML models should be added to text.

-          Future work is missing.

Reviewer 3 Report

Comments and Suggestions for Authors

This paper proposed a gait recognition method by using data from five IMUs sensors fixed on the soft exoskeleton. It identified walk styles with the algorithm of support vector machine (SVM) and gait phases with long and short-term memory (LSTM). The paper is very well written. The experimental part is approved by the Ethics advisory committee.

Round 2

Reviewer 1 Report

Comments and Suggestions for Authors

All my concerns have been adequately addressed in the revised version. Thanks to the authors for improving the paper.

Comments on the Quality of English Language

NA

Reviewer 2 Report

Comments and Suggestions for Authors

The authors answered my concerns and revised accordingly.